# Impacts of shared mobility on vehicle lifetimes and on the carbon footprint of electric vehicles

Johannes Morfeldt [1] ✉ & Daniel J. A. Johansson [1]

Shared cars will likely have larger annual vehicle driving distances than individually owned cars. This may accelerate passenger car retirement. Here we develop a semi-empirical lifetime-driving intensity model using statistics on Swedish vehicle retirement. This semi-empirical model is integrated with a carbon footprint model, which considers future decarbonization pathways. In this work, we show that the carbon footprint depends on the cumulative driving distance, which depends on both driving intensity and calendar aging. Higher driving intensities generally result in lower carbon footprints due to increased cumulative driving distance over the vehicle's lifetime. Shared cars could decrease the carbon footprint by about 41% in 2050, if one shared vehicle replaces ten individually owned vehicles. However, potential empty travel by autonomous shared vehicles—the additional distance traveled to pick up passengers—may cause carbon footprints to increase. Hence, vehicle durability and empty travel should be considered when designing low-carbon car sharing systems.

Decarbonizing road transportation is an important step in achieving the Paris Agreement[1], with battery electric vehicles (BEVs) being one of the main strategies considered[2,3]. Transitioning towards a fully electrified passenger car fleet effectively eliminates tailpipe carbon dioxide ($CO_2$) emissions and has the potential to significantly reduce lifecycle $CO_2$ emissions[4]. Nevertheless, social and environmental sustainability concerns have been raised related to battery manufacturing and the mining of raw materials[5].

Pathways with low resource exploitation and high energy efficiency are beneficial for decarbonization since they reduce the overall energy demand and material requirements. Options for passenger cars include various on-demand mobility schemes (including ride sourcing and ride sharing) that could replace individual passenger car ownership[6–8]. Implementing such schemes on a large scale would probably depend on self-driving (autonomous) vehicles[9,10]. Autonomous vehicles could decrease costs and increase the convenience of such schemes thus rendering it preferable over individually owned cars (including other arrangements where the car is primarily used by one household)[9].

Car sharing and ride sharing could increase resource efficiency and reduce the environmental load of the system by replacing on the order of ten individually owned cars per shared car[11]. At the same time, the shared cars will likely be used more intensively during their lifetimes as compared to individually owned cars[12]. Moreover, shared autonomous vehicles may travel around without passengers for a large extent of their cumulative lifetime distance (so called "empty travel" or "deadhead travel"), which could lead to faster vehicle fleet turnover and increased manufacturing-phase emissions[13]. Simulation studies of shared autonomous vehicles have found that empty travel could increase the total vehicle travel distance by 10 to 100% in urban areas compared to the intended travel distance (i.e., the distance traveled by the car to transport a passenger or a group of passenger from one point to another)[11,14]. Empirical studies show a level of around 60% for taxi rides[15] and 40% for ride-sourcing services[16,17] on top of the intended (or served) travel distance.

The cumulative driving distance over the vehicle's lifetime is an important assumption when estimating the carbon footprint of passenger car travel but varies significantly among studies[18] and has been

[1]Physical Resource Theory, Department of Space, Earth and Environment, Chalmers University of Technology, Maskingränd 2, SE-412 96 Gothenburg, Sweden. ✉e-mail: johannes.morfeldt@chalmers.se

shown to have large impacts on the results[19]. This assumption is even more uncertain for future mobility schemes, including systems based on car sharing or ride sharing[20,21]. Nevertheless, carbon footprint studies tend to assume that shared autonomous BEVs would travel at least as far as current taxis over the course of their lifetimes[12,13,22,23]. Hence, considering a relationship between driving intensity and vehicle lifetime is critical when assessing the carbon footprint of shared autonomous BEVs.

Studies using survival analysis[24,25] have determined that both calendar age and cumulative driving distance are important for the decision to retire a vehicle. Studies using statistical analyses of historical data have also shown that changes in driving intensity over the lifetime of the vehicle can have impacts on $CO_2$ emissions[26] and that vehicle lifetime extensions can result in lower carbon footprints[27]. However, to our knowledge, no study has yet attempted to establish a relationship between driving intensity and vehicle lifetime, and the implications of such a relationship on carbon footprints. Moreover, the carbon footprint-related consequences of changes in driving intensity in response to shared autonomous BEVs and plausible levels of empty travel have not yet been analyzed in situations where energy systems are decarbonized over time. To meet the goals of the Paris Agreement, shifts towards low-carbon manufacturing processes and electricity mixes used for charging needs to happen over the course of the next 30 years[2,4]. Thus, the vehicle's lifetime, its annual driving intensity, and its interaction with decarbonizing energy systems will play important roles for the carbon footprint of passenger car travel over the coming decades.

In this work, we aim to bridge this research gap by estimating the impact of vehicle lifetime and annual driving intensity on the carbon footprints of passenger cars used for sharing. We design a semi-empirical lifetime-intensity model for assessing the lifetime of passenger cars with increasing annual driving intensity. The model is used together with prospective lifecycle assessment using vehicle fleet turnover simulations to assess the carbon footprint impacts of shared autonomous BEVs and potential levels of empty travel. The effects of climate change mitigation in global vehicle manufacturing and electricity generation are considered in the assessment. These energy and industrial systems are assumed to decarbonize in line with the Paris Agreement's goals for the results presented in the main article, while results for an alternative pathway in line with currently stated policies are presented in the Supplementary Information. We show that the carbon footprint depends on the cumulative driving distance, which depends on both driving intensity and calendar aging. Higher driving intensities generally result in lower carbon footprints due to increased cumulative driving distance over the vehicle's lifetime. Shared cars could decrease the carbon footprint by about 41% in 2050, if one shared vehicle replaces ten individually owned vehicles. However, potential empty travel by autonomous shared vehicles—the additional distance traveled to pick up passengers—may cause carbon footprints to increase. Hence, vehicles should be designed for durability, and empty travel should be kept low, to enhance the carbon footprint benefits of sharing.

## Results

### Vehicle lifetimes decrease with increased driving intensity

Statistics on vehicle retirement can provide insights into how vehicle lifetimes vary with driving intensity. Most Swedish vehicles retired in 2014–2018 had a lifetime between 7 and 26 years and lifetime driving distances between 43 and 390 thousand kilometers (km) (95% interval). Calculating the average annual driving intensity for these vehicles results in a range between 0.5 and 28 thousand km per year (95% interval). All vehicles analyzed are internal combustion engine vehicles (ICEVs) since we are interested in capturing the behavior of mature vehicle technologies; very few BEVs have been retired so far.

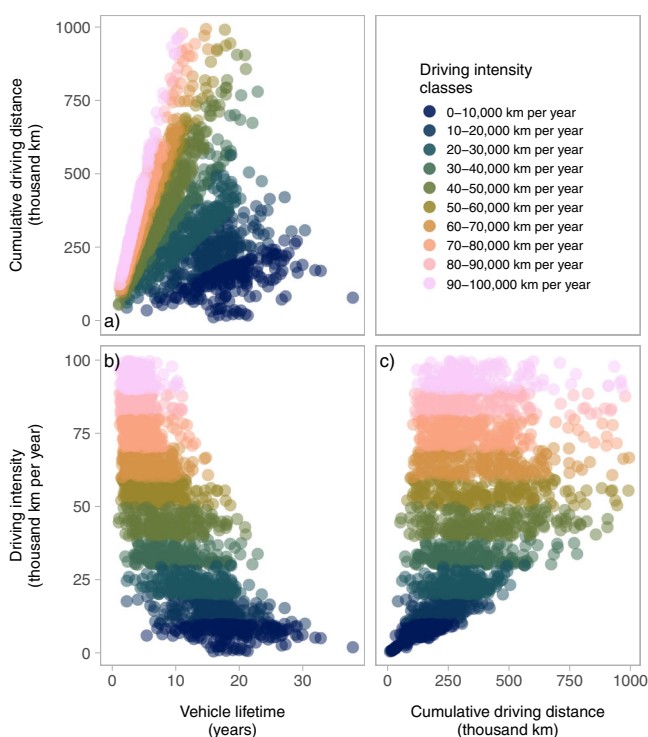

**Fig. 1 | Statistical analysis of vehicle retirements. a** Vehicle lifetime and cumulative driving distance. **b** Vehicle lifetime and average driving intensity. **c** Cumulative driving distance and average driving intensity. Results are shown for stratified samples based on average annual driving intensity classes of Swedish ICEVs retired between 2014 and 2018. The color indicates the driving intensity class of the data point.

The statistics show an average lifetime of 16.3 years, average lifetime driving distance of 216 thousand km, and an average annual driving intensity of 14.2 thousand km per year. Note that while a normal distribution can approximate vehicle lifetimes well, lifetime distances may be better approximated by a Weibull distribution, see Supplementary Figs. 2–4, confirming previous research[14]. Since the sample is unevenly distributed over driving intensities with a bias towards the mean, stratification is used as a starting point for characterizing how the vehicle lifetimes vary with average annual driving intensity, see Fig. 1 and details on the stratified samples in Supplementary Table 4.

The stratification is made for individual average annual driving intensity classes, varying from 0 to 100,000 km per year in steps of 10,000 km per year. For each individual driving intensity class, a close to linear relationship exists between vehicle lifetime and cumulative driving distance. The linear slope becomes steeper with each higher driving intensity class, see Fig. 1a. This suggests that the calendar age of a vehicle becomes generally shorter with increasing annual driving intensity. Further, the cumulative driving distances are distributed across a wide range for higher driving intensity classes, see Fig. 1c, while the distribution is narrower for lower driving intensities. Hence, the probability distribution of retirement becomes wider as the annual driving intensity increases, which means that the probability of a retirement decision at a specific cumulative driving distance becomes smaller. Finally, the distribution of vehicle lifetimes becomes narrower and shifts towards lower vehicle lifetimes as the average driving intensity increases, see Fig. 1b. Hence, we focus the following analysis on empirically describing the relationship between driving intensity and vehicle lifetime in order to capture the impact of vehicle use on retirement age. The data presented here does not corroborate the assumption of a fixed cumulative driving distance, which is assumed in many lifecycle assessments of vehicles[13,18].

The average vehicle lifetime decreases with each higher driving intensity class, from 19 years for average driving intensities of 0–10,000 km per year to 3.9 years for average driving intensities of 90,001–100,000 km per year, see Fig. 1b. The standard deviation of the distributions also indicates that the range of probable lifetimes becomes narrower with increasing annual driving intensity (although the standard deviation increases in relative terms). The standard deviation decreases from 5.0 years for driving intensities of 0–10,000 km per year to 1.9 years for driving intensities of 90,001–100,000 km per year. Results for a categorization in four vehicle sizes (mini, medium, large, and luxury size cars, see Supplementary Fig. 5) suggest that cars with low annual driving intensity are mainly represented by small size cars, while large to luxury size cars mainly have higher annual driving intensities. Medium size cars cover the full spectrum of annual driving intensities.

Currently, battery degradation is often raised as a constrain to the cumulative driving distance and lifetime of BEVs[28–30], but the BEV is a relatively new technology on the market and, hence, statistics on battery lifetimes from real-world driving are scarce. The number of electric vehicles on the world's roads were in the thousands in 2010 and grew rapidly to reach about 2 million by 2016 and over 10 million by 2020[31,32]. Hence, if enough retirement statistics for electric vehicles were available to make thorough statistical analyses, most vehicles would be much less than 10 years old. However, the limited data currently available on cars with batteries in Swedish vehicle retirement statistics show similar distributions as the stratified data presented above, see Supplementary Notes 1–3 and Supplementary Figs. 11, 12. The data show shorter lifetimes on average (due to the limited historic data on electrified vehicles) and with a bias towards hybrid electric vehicles (HEVs) due to very few BEVs and plug-in hybrid electric vehicles (PHEVs) having been retired during the analyzed period.

Many BEV manufacturers already have warranties for their batteries of about seven to eight years or about 150,000 to 240,000 km, whichever comes first[33–37]. Future battery chemistries may further reduce degradation. Some studies suggest that future batteries may have significantly longer lifetimes than today. This is expected in response to altered battery chemistries[38], changes in charging and use behavior[39], and/or changed battery design[40]. Those changes could potentially yield a cumulative driving distance of more than three million kilometers—effectively outliving the rest of the vehicle. These improvements, if they materialize, would likely improve the cycling of the batteries. However, other factors could still limit the vehicle's lifetime[25], such as accidents, aging of other vehicle parts (e.g., structural elements of chassis and body), economic reasons, and consumer trends. Further, the durability of the vehicle is significantly dependent on the vehicle design, material selection, and business models[41].

In summary, the results suggest that the annual driving intensity indeed has a strong influence on vehicle lifetimes. The relationship between driving intensity and vehicle lifetime may differ between BEVs and ICEVs, but not enough data is yet available to make such an analysis thoroughly. As a consequence, the remainder of this article explores how changes in annual driving intensity may influence the carbon footprint of passenger car travel, assuming that the relationship shown for ICEVs is applicable as a proxy for individually owned and shared autonomous BEVs. We capture the uncertainty in future vehicle lifetimes of (shared and autonomous) BEVs by highlighting extreme values for the relationship between annual driving intensity and vehicles lifetime as well as the empirically estimated relationship based on ICEV retirement data.

## Impact of driving intensity on fleet-wide carbon footprints

This section presents carbon footprint estimates for BEVs at different average annual driving intensities based on the developed semi-empirical lifetime-intensity model, see Methods section for full description and discussion of the design. The model estimates the

expected lifetime of a vehicle given a certain assumed average annual driving intensity. As we discussed in the previous section and in Supplementary Notes 1–3, it is assumed that the lifetime-intensity model is representative for BEVs despite being calibrated on data for ICEVs. Note also that we assume that current average vehicles in terms of weight are representative for future systems[4].

To capture the relationship between driving intensity and vehicle lifetime, we use the elasticity design of the lifetime-intensity model with Weibull distribution, see Eqs. (2), (3), (6) in Methods section, and elasticities ($\varepsilon \approx -0.65$ and $\beta \approx 0.51$) in the simulations. The lifetime-intensity model is trained with empirical data (i.e., Swedish vehicle retirement statistics described in the previous section) using maximum likelihood estimation, see Supplementary Tables 6, 7. A lifetime-intensity elasticity of −0.65 implies that the scale of the lifetime, is reduced by about −0.65% if annual driving intensity is increased by 1%. The scale parameter can be seen as the characteristic lifetime and is close to the average lifetime. Consequently, the average cumulative lifetime driving distance increases by approximately 0.35% on average if annual driving intensity is increased by 1%.

Carbon footprints are also estimated for two extreme cases, $\varepsilon = 0$ and $\varepsilon = -1$, representing no influence of driving intensity on lifetime and full influence of driving intensity, respectively. The two extreme cases show the sensitivity of the model design to the assumed elasticity. The range represents possible cases if the model was trained on different retirement data. $\varepsilon = 0$ is a relevant extreme case if future individually owned and/or shared autonomous BEVs would age in a way where driving intensity has no importance in the decision to retire vehicles. This could be the case if the vehicle and battery degradation are only influenced by calendar age. $\varepsilon = -1$ represents a case where vehicle aging, including aging of the battery, is only dependent on the distance driven (e.g., if battery aging only depends on the number of charging cycles). This latter approach is used in many lifecycle assessments[13,18], where fixed cumulative vehicle distances are assumed. Note though that $\beta$ is based on the empirical data ($\beta \approx 0.51$) also for the extreme cases.

The impact of driving intensity on the carbon footprint of BEVs is estimated using a vehicle fleet turnover simulation set to meet a certain annual travel demand. Hence, fewer cars are needed to meet the travel demand if the average annual driving intensity per vehicle increases, see details in Methods. The carbon footprint is presented as emissions per vehicle-kilometer, based on the average annual emissions for a given year, including emissions from electricity used for charging and vehicle manufacturing, divided by the travel demand of that year. Figure 2 shows the results for BEVs assuming global electricity technology mix and that global manufacturing and electricity generation follow a climate change mitigation pathway in line with the goals of the Paris Agreement. The assumed pathways for carbon intensities of electricity generation used for charging are shown in Supplementary Fig. 1.

Emissions per vehicle-kilometer related to vehicle manufacturing decrease with increasing driving intensity in all cases, see Fig. 2b, e. The reason is that increased average annual driving intensity results in that fewer cars are needed to meet the travel demand. Emissions per vehicle-kilometer in the use-phase are constant for all cases since total use-phase emissions are proportionate to the travel demand, see Fig. 2c, f. Intuitively, average use-phase emissions depend on the vehicle-specific energy use and the carbon intensity of the electricity mix used for charging in each specific year, which can be seen when comparing the level in Fig. 2c, f. Hence, the total emissions per vehicle-kilometer varies with manufacturing emissions when increasing the driving intensity, see Fig. 2a, d.

As expected, manufacturing-phase emissions decrease rapidly and approach zero with increasing driving intensity when vehicle retirement is unaffected by driving intensity, i.e., $\varepsilon = 0$, displayed as dashed lines in Fig. 2. This is due to the retirement decision solely

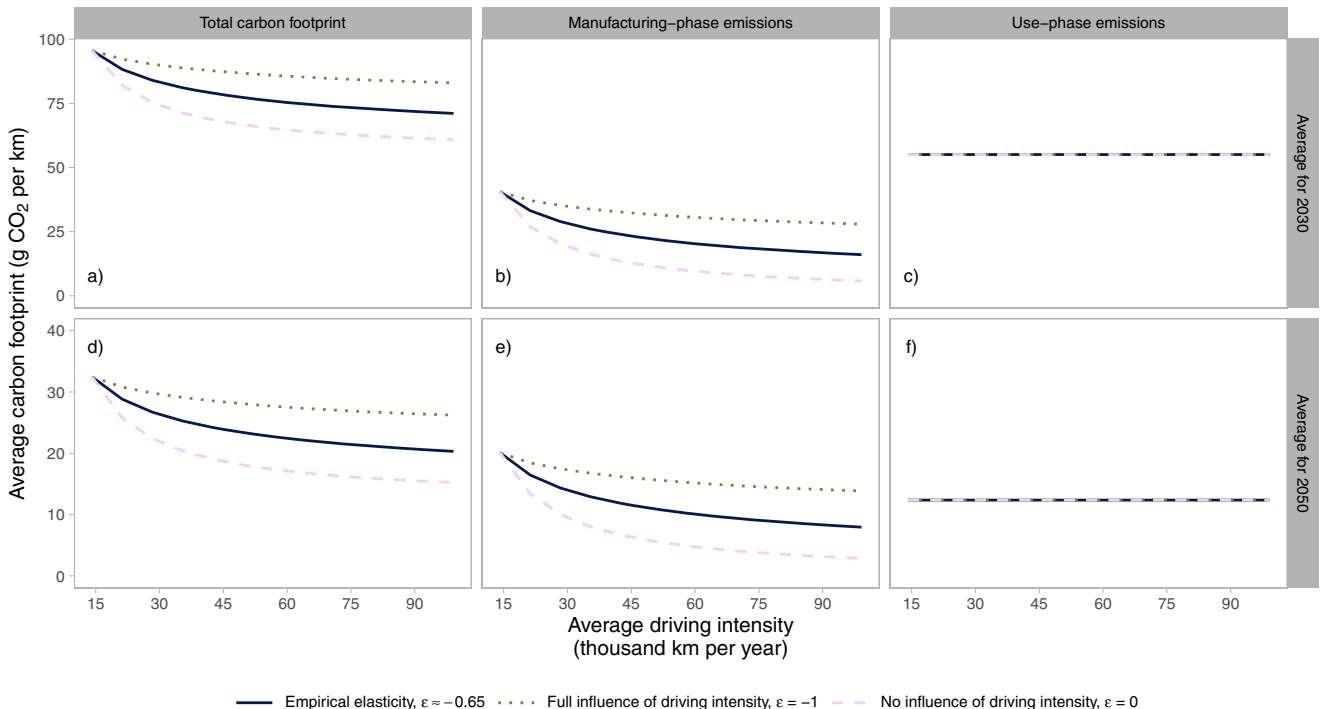

**Fig. 2 | Impact of lifetime-intensity model on carbon footprint.** Results show the impact on annual average carbon footprints for meeting a certain travel demand: total carbon footprint (**a**, **d**), manufacturing-phase emissions (**b**, **e**), and use-phase emissions (**c**, **f**) for 2030 (**a**–**c**) and for 2050 (**d**–**f**), depending on the elasticity of the semi-empirical lifetime-intensity model (solid: empirical elasticity, $\varepsilon \approx -0.65$, dotted: full influence of driving intensity, $\varepsilon = -1$, and dashed: no influence of driving intensity, $\varepsilon = 0$). The results assume that global manufacturing and electricity generation decarbonize in line with the Paris Agreement's goals.

depending on the calendar age when $\varepsilon = 0$, which is based on the empirical calendar age distribution in the model. This can be compared to when vehicle retirement is largely affected by the driving intensity, i.e., $\varepsilon = -1$, displayed as dotted lines in Fig. 2. In this case, the driving distance over the whole lifetime of each vehicle is close to independent of the driving intensity. Hence, the reduction in the number of cars needed to meet the travel demand when the annual driving intensity increases would be counteracted by the number of retired vehicles that reach their maximum cumulative driving distance. This results in an inflow of new vehicles needed to replace the retired ones, which is close to independent of the driving intensity. The reasons that the number of vehicles slightly drop with increasing intensity when $\varepsilon = -1$ are the characteristics of the Weibull distribution, how it shifts as the intensity increases, and that the annual driving intensity for each individual car is assumed to drop by 4.4% per year. The significance of the drop in the annual driving intensity is tested in Supplementary Fig. 9, showing that manufacturing emissions decrease less when the driving intensity is assumed to be constant over the lifetime of each vehicle.

A lifetime-intensity elasticity based on empirical evidence, i.e., $\varepsilon = -0.65$, results in a development in-between the two extremes, displayed as solid lines in Fig. 2. A sensitivity analysis shows that the shape of the curves for total carbon footprint are scaled but similar in relative terms when assuming average Swedish or European Union (EU) electricity for charging, see Supplementary Fig. 9. Further, the general pattern of the relationship between average annual driving intensity and the carbon footprint is similar also if energy systems and global manufacturing do not decarbonize in line with the Paris Agreement and instead develops according to stated policies, see Supplementary Fig. 9.

To summarize, our results show that measures intended to increase annual driving intensity of individual cars to meet a given travel demand would result in carbon footprint reductions. Such measures include car sharing services, e.g., existing ride sourcing

systems and future systems using shared autonomous BEVs. Such services could replace individual car ownership, but they may also increase driving distances because of empty travel to pick up passengers. This risk is evident for current taxis and ride sourcing services[15–17] as well as in simulations of future transport systems using shared autonomous vehicles[11,14]. In the next section, we explore how empty travel could impact the carbon footprint when simultaneously considering the possible influence that increased driving intensity might have on the lifetime of vehicles.

**Empty travel by autonomous vehicles may increase emissions**
The risk of empty travel when using on-demand mobility services, including those provided by autonomous vehicles, could reduce the resource and environmental efficiency of sharing. The lifetime-intensity model shows that the lifetime of the vehicle is likely to decrease with increased annual driving intensity. Vehicles may need to be replaced more often if a large part of that annual driving intensity is made up by empty travel, with increasing emissions in vehicle and battery manufacturing as a result[13]. Here, we explore how the carbon footprints of individually owned BEVs, individually owned autonomous BEVs and shared autonomous BEVs depend on the elasticity of the lifetime-intensity model and the share of empty travel. Further, we estimate the breakeven level of empty travel, i.e., the point where the carbon footprints of a fleet of shared autonomous BEVs and one of individually owned BEVs (without any empty travel) are equal.

The impact of empty travel on the carbon footprint for a fleet of shared autonomous BEVs is analyzed using the vehicle fleet turnover simulation, see details in Methods section. Simulations are made for assumptions on additional empty travel on top of the intended travel—ranging from 0 to 100%, and for assumptions on how many individually owned BEVs a shared autonomous BEV replaces—five or ten, while still meeting the given level of annual travel demand. Specific energy use per km is assumed to be the same irrespective of if the car is autonomous or not. Note though that the combination of a shared autonomous BEV

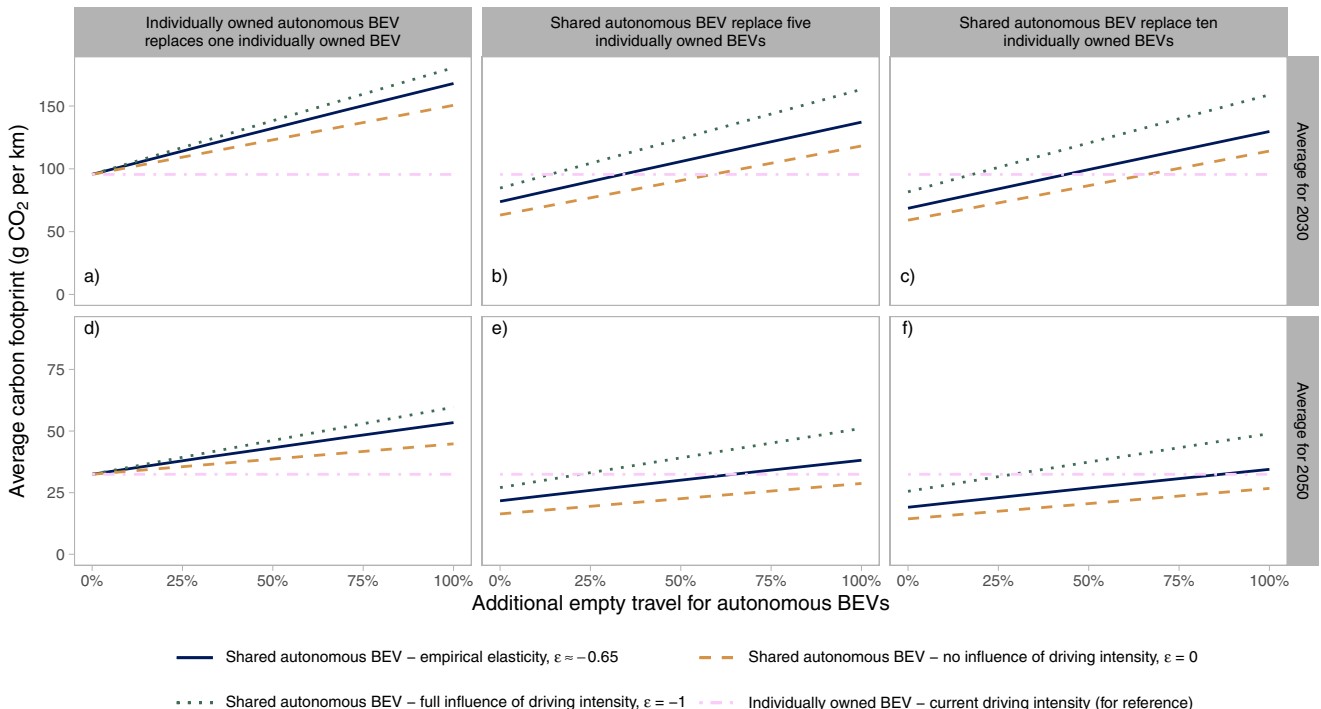

**Fig. 3 | Breakeven level of empty travel.** Estimated based on the average carbon footprint of individually owned autonomous BEVs that replace one individually owned BEV (**a**, **d**) and shared autonomous BEVs replacing five (**b**, **e**) or ten (**c**, **f**) individually owned BEVs depending on the elasticity of the semi-empirical lifetime-intensity model (solid: shared autonomous BEV – empirical elasticity, $\varepsilon \approx -0.65$, dotted: shared autonomous BEV – full influence of driving intensity, $\varepsilon = -1$, dashed: shared autonomous BEV – no influence of driving intensity, $\varepsilon = 0$, and dot-dashed: individually owned BEV – current driving intensity). **a–c** Show results for 2030 and **d–f** show results for 2050. The results assume that global manufacturing and electricity generation decarbonize in line with the Paris Agreement's goals.

replacing ten individually owned BEVs and assuming 100% empty travel results in high annual driving intensity of ca 280,000 km, which may not be possible to achieve for one car. Hence, such extreme combinations are included only for illustrative purposes. We also analyze one case with individually owned autonomous BEVs that are not shared but still may travel empty. This can occur, for example, when autonomously parking and/or charging at remote spots.

In the case where an individually owned autonomous BEV causes empty travel, the breakeven level occurs at 0% as expected, see Fig. 3a, d. This means that any empty travel caused by using the autonomous BEV results in an increase in the average carbon footprint, as compared to using a regular BEV to meet the same travel demand. In the case where individually owned BEVs are replaced by shared autonomous BEVs, we first note that a system with shared autonomous BEVs in 2030 reduces the carbon footprint per intended km traveled if no empty travel is assumed. The carbon footprint decreases from 96 g $CO_2$ per km for individually owned BEVs to 74 and 69 g $CO_2$ per km if one shared autonomous BEV replaces five or ten individually owned BEVs, respectively, assuming empirical elasticity for the lifetime-intensity model. The corresponding numbers for 2050 are 32 g $CO_2$ per km for individually owned BEVs, and 22 and 19 g $CO_2$ per km if one shared autonomous BEV replaces five or ten individually owned BEVs, respectively.

The breakeven level of empty travel for a fleet in 2030 occurs at 34 and 44% for shared autonomous BEVs that replace five and ten individually owned BEVs, respectively, see intersection between solid and dot-dashed lines in Fig. 3b, c. As global manufacturing and electricity generation decarbonize further, additional levels of empty travel are possible before breakeven with the carbon footprint of individually owned BEVs is reached. Hence, for a fleet in 2050, the breakeven level of empty travel increases to 64% and 87% for shared autonomous BEVs that replace five and ten individually owned BEVs, respectively, Fig. 3e, f.

As discussed in the previous section, only manufacturing-phase emissions are affected by the lifetime-intensity model. A larger negative elasticity implies a larger inflow and outflow of shared autonomous BEVs in each year, implying a larger average carbon footprint, see Supplementary Fig. 8. The elasticity representing no influence of driving intensity on vehicle lifetime ($\varepsilon = 0$) results in higher breakeven levels as compared to the elasticity based on empirical evidence. In this case, for 2030, the breakeven level is 59 and 66% for shared autonomous BEVs that replace five and ten BEVs, respectively, see dashed lines in Fig. 3b, c, and over 100% in 2050, see dashed lines in Fig. 3e, f. Conversely, the elasticity representing full influence of driving intensity on vehicle lifetime ($\varepsilon = -1$) results in lower breakeven levels. In this case for 2030, 14 and 18% for shared autonomous BEVs that replace five and ten individually owned BEVs, respectively, see dotted lines in Fig. 3b, c, and 22 and 29% for 2050, respectively, see dotted lines in Fig. 3e, f.

A sensitivity analysis shows lower breakeven levels if global manufacturing and electricity generation follows a pathway in line with stated policies rather than one that achieves the goals of the Paris Agreement, see Supplementary Fig. 10. It also shows that the breakeven level is significantly higher, above 100% in several cases, if lower carbon intensities are assumed for electricity used for charging (i.e., Swedish or European average electricity). The sensitivity of the assumed driving intensity decrease rate of 4.4% is also tested, showing higher breakeven levels for the additional empty travel with higher driving intensity decrease rates (i.e., when a larger share of the travel for one vehicle is concentrated to early years in the vehicle's lifetime), while the opposite holds if the driving intensity is constant over time. Nevertheless, the assumed elasticity in the lifetime-intensity model has a higher impact on the results than the assumed driving intensity decrease rate.

The significance of the elasticity in these results points to the importance of designing future shared autonomous BEVs for

durability. The reason for this can be summarized: the smaller the reduction in lifetime when increasing driving intensity, the larger the potential carbon footprint benefits of car sharing.

## Discussion

The passenger transport systems are likely to go through several changes during the coming decades. The most prominent changes include increased use of electrified and autonomous vehicles as well as on-demand mobility schemes, including car sharing and ride sharing. These trends will affect the pathways towards decarbonization of passenger car travel, including changes in charging patterns[13], cost structures[9], and the value of travel time[42–44], which may induce additional travel activity[45] and cause modal shifts[46,47]. These trends may also cause changes in vehicle design, including materials used in manufacturing[48] and changes to facilitate material recycling[49], but many of these aspects are yet to materialize.

Our analysis shows that the relationship between vehicle lifetime and driving intensity is an important factor when estimating the carbon footprint of shared mobility. Some analysts argue that passenger cars in today's fleets are not being used enough to compensate for material use and emissions during the manufacturing phase[49,50]. Therefore, increasing the driving intensity, for example through shared autonomous BEVs, may be an option for reducing lifecycle emissions from passenger car travel. However, if increasing driving intensity also results in shortened vehicle lifetimes, as suggested by the statistics, the carbon footprint would not drop as much as if a fixed lifetime were assumed.

The statistical analysis and the results from the designed semi-empirical lifetime-intensity model suggest that increased intensity of vehicle use tends to increase the cumulative lifetime distance. Hence, the results indicate that shared autonomous BEVs would reduce the carbon footprint if it results in higher driving intensity of each individual vehicle. For example, we find that a system with shared autonomous BEVs can decrease the carbon footprint per kilometer of intended travel by about 41% if one shared vehicle replaces 10 individually owned vehicles in 2050. However, this assumes a level of zero empty travel. We show that the potential carbon footprint benefit can be reduced—and even erased—if the level of empty travel becomes large. Further, besides avoiding excessive empty travel, the emissions reduction potential of shared mobility could be further improved if ride sharing is adopted, since each traveler sharing the ride in that case would bear part of the carbon footprint by effectively increasing the occupancy ratio. Note that induced travel by autonomous BEVs (both individually owned and shared) has not been assessed in this study. Nevertheless, this risk is important to consider since the use of autonomous vehicles may substantially increase the travel demand. Autonomous vehicles may effectively reduce the value of travel time and the generalized travel cost[45] since the driver does not need to be attentive and can instead use their time in the vehicle for whatever they find convenient. The potential increase in the travel demand that may follow from reduced value of travel time could further increase the total carbon footprint for the fleet as a whole.

Finally, our conclusions rely on the assumption that the relationship between driving intensity and vehicle lifetime established in the semi-empirical model will hold also for future regular and autonomous BEVs. In this article, we present preliminary evidence suggesting that cars with batteries follow similar trends as ICEVs, but the design and lifetimes of future batteries and vehicles are highly uncertain. Hence, the intention with the analysis presented in this paper is to highlight potential consequences based on currently available data and discuss them in relation to alternative assumptions. Those alternative assumptions highlight a range of plausible outcomes if the lifetime characteristics of future batteries and vehicles may deviate from those of current passenger cars. Nevertheless, the analysis shows that the carbon footprint may be substantially reduced if the relationship

between average annual driving intensity and vehicle lifetime is weakened, pointing to the importance of designing future BEVs (both autonomous and regular) for durability.

## Methods

### Swedish vehicle retirement statistics

Statistics on Swedish passenger cars retired between 2014 and 2018 are used to understand how changes in annual average driving intensity could influence vehicle lifetimes. The statistics are collected from the Swedish registry for road transport vehicles, regulated by Swedish law[51]. The excerpt, provided by the Swedish government agency Transport Analysis[52], includes information on the manufacturing year, date of registration, car manufacturer, engine type, mass in running order, cumulative distance traveled at last inspection, date of last inspection, and date of deregistration. The excerpt only includes vehicles that were indeed retired at the date of deregistration. Hence, vehicles that were deregistered for administrative reasons or exported are excluded.

The filtered dataset includes 442,395 observations. The filtering performed by the authors aims to reduce bias in the results and applies the following criteria: (i) age or distance traveled must not be missing, equal to zero, or equal to 999,999, (ii) time between last inspection and date of deregistration must not be longer than 14 months, (iii) time between first registration of the vehicle and the manufacturing year must not be longer than two years, (iv) average distance traveled must not be greater than 600 km per day, (v) average distance traveled must not be less than 1 km per day, (vi) mass in running order must not be greater than 3000 kg, and (vii) engine type is gasoline or diesel without hybridization, ethanol or natural gas/biogas. Details and rationale for these criteria are provided in Supplementary Tables 1–3. Criterion (ii) filters many observations but including them does not significantly impact the results.

Stratified random sampling is used to create a new dataset for analyzing the influence of increasing driving intensity since only a small share of the dataset represents cars with high average annual driving intensity, such as taxis or other commercial vehicles. The strata and random sample size are set to maximize the amount of information about vehicles with high driving intensity while also ensuring high enough sample size to enable further statistical analysis. This results in strata for average annual driving intensity classes of 10,000 km/year increments from 0 km/year to 100,000 km/year. The random sample size in each stratum is 200 observations, except for the highest intensity class where the whole sample of 145 observations is used, see Supplementary Table 4.

### Semi-empirical lifetime-intensity model

The semi-empirical lifetime-intensity model enables estimations of vehicle lifetime probabilities for a given annual average driving intensity. The model can easily be updated with new parameters on average vehicle retirement lifetime, its standard deviation, and the average annual driving distance, as new statistics become available. The model can also easily be recalibrated based on new stratified random sampling datasets to enable use for other geographical regions. Two model designs are considered together with two assumptions on the probability distribution of the lifetime data as a result of these prerequisites.

If the data is assumed to follow a normal distribution, we assume that the probability of a vehicle manufactured at year $t_0$, with average annual driving intensity $D$, being retired at year $t$ is

$$\Phi_n(t, t_0, D) = \int_{t_0}^{t} \frac{1}{\sigma(D)\sqrt{2\pi}} e^{-\frac{1}{2}\left(\frac{(t-t_0)-\mu(D)}{\sigma(D)}\right)^2} dt. \tag{1}$$

In the elasticity design, we introduce a factor dependent on the quota between the annual driving intensity of the vehicle and the

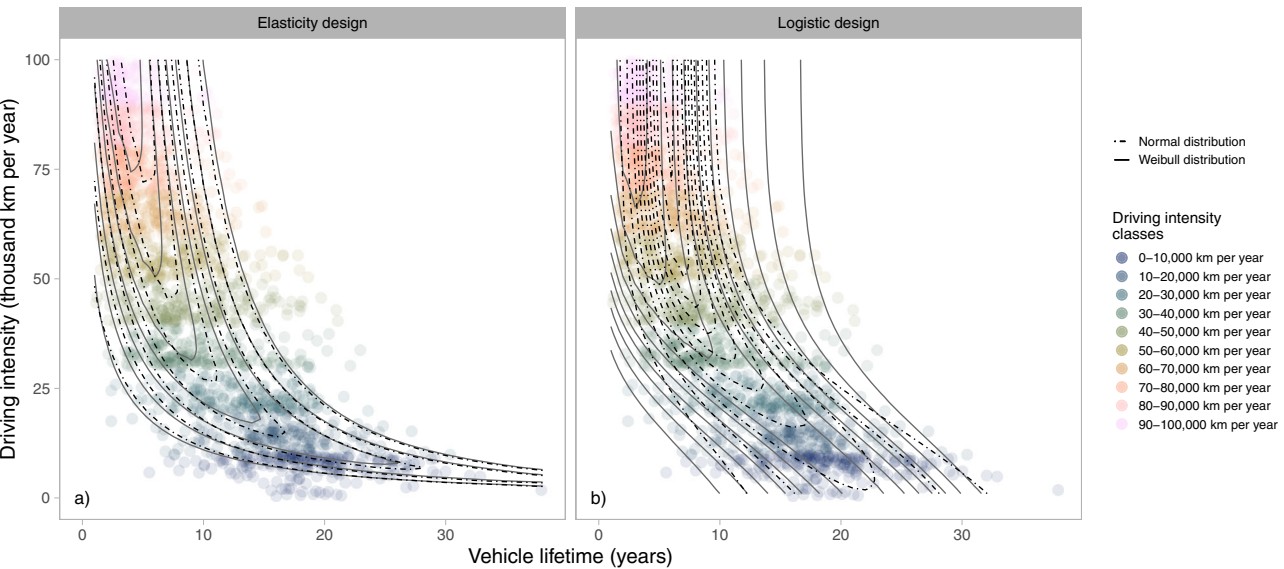

**Fig. 4 | Semi-empirical model results. a** Results for the elasticity design. **b** Results for the logistic design. The contours show probability density levels and are provided for both Normal (dot-dashed) and Weibull distributions (solid). Stratified samples of Swedish vehicle retirement statistics for 2014–2018 are provided in the background for comparison. The color indicates the driving intensity class of the data point.

average annual driving intensity of current vehicle retirements, $D_0$, as part of the mean,

$$\mu(D) = \tau_0 \left(\frac{D}{D_0}\right)^{\varepsilon} \tag{2}$$

that adjusts the expected vehicle lifetime of current retirements, $\tau_0$, dependent on the elasticity, $\varepsilon$, that decides the level of influence of the driving intensity. An elasticity of −1 implies that the vehicle lifetime is fully determined by the driving intensity (e.g., if driving intensity is doubled, lifetime is halved), 0 indicates no influence and the lifetime is only determined by calendar age, while an elasticity above 0 would imply that the vehicle lifetime increases with driving intensity. This design benefits from easy interpretation, but it only applies for driving intensities equal to or greater than the current average, see Fig. 4.

The standard deviation,

$$\sigma(D) = \alpha \tau_0 \left(\frac{D}{D_0}\right)^{\varepsilon\beta} \tag{3}$$

is designed in a similar way to the design for the mean, where the constant $\alpha = \frac{\sigma_0}{\tau_0}$ is determined based on a fit of a normal distribution to current vehicle retirement statistics. An additional elasticity, $\beta$, is introduced in the standard deviation to account for the distributions becoming increasingly narrow with higher driving intensity classes, see Fig. 1b.

In the logistic design, we instead assume that the distribution is governed by a function inspired by the logistic curve to better capture the form of the stratified random sampling. The logistic curve function is slightly altered to reduce the number of parameters to fit to the data. Hence, $\mu(D)$ and $\sigma(D)$ are defined as follows in this design.

$$\mu(D) = L_0 - \frac{L}{1 + e^{(1-D/D_0)}} \tag{4}$$

and

$$\sigma(D) = \alpha \left(L_0 - \frac{L}{1 + e^{(1-D/D_0)}}\right), \tag{5}$$

where $L$ and $L_0$ are the parameters that would be calibrated based on the stratified random sampling. This design applies for all driving intensities greater than zero.

If the data are assumed to follow a Weibull distribution, we assume that the probability of a vehicle manufactured in year $t_0$, with average annual driving intensity $D$, being retired at year $t$, is

$$\Phi_W(t,t_0,D) = \int_{t_0}^{t} \frac{k(D)}{\lambda(D)} \left(\frac{(t-t_0)}{\lambda(D)}\right)^{(k(D)-1)} e^{-\left(\frac{(t-t_0)}{\lambda(D)}\right)^{k(D)}} dt, \tag{6}$$

where the scale, $\lambda(D)$, is defined in the same way as the mean, $\mu(D)$, see Eqs. (2, 4) above, and shape, $k(D)$, is defined in the same way as the standard deviation, $\sigma(D)$, see Eqs. (3, 5) above. Note that $\tau_0$ in this case represents the scale of current vehicle retirement statistics and that $\alpha = \frac{k_0}{\tau_0}$ is determined by fitting a Weibull distribution. The fact that the median is lower than the mean for higher driving intensity classes indicates that the distribution is more positively skewed for higher driving intensity classes. This suggests that a Weibull distribution with a longer tail towards higher vehicle lifetimes would be a better fit, confirming previous research[24,53].

The parameters for the different model designs are estimated using maximum likelihood estimation, see Supplementary Table 7. A comparison of modeled vehicle lifetimes with the stratified random samples for different driving intensity classes is presented in Fig. 4 and Supplementary Fig. 7. The contour lines in Fig. 4, also known as iso-density lines[54], show how the points of equal probability density for a given vehicle lifetime shift depending on the assumed driving intensity (y-axis) and on the model design (panel and line type). The highest probability density level is shown around the mean of the distribution, and the distance indicates the rate of change. This implies that a larger distance between the lines indicates a more spread-out distribution, analogously to on a topographic map.

Figure 4a clearly shows that the elasticity design deviates from the statistics at the average current driving intensity of 14,200 km per year and approaches an infinite lifetime as driving intensities decrease. The proposed correction for this issue is to use the logistic design, as demonstrated in Fig. 4b. However, a limitation of the logistic design is that the distribution of vehicle lifetimes is assumed to be kept constant for driving intensities higher than the stratum with highest driving intensity (i.e., higher than 100,000 km per year in this study), see

**Table 1 | Benefits and drawbacks different semi-empirical model designs**

| Design | Benefits | Drawbacks |
|---|---|---|
| Elasticity model | • Simple formulation<br>• Easy to interpret | • Applies to driving intensities equal to or greater than current average |
| Logistic model | • Applies for all driving intensities | • Less intuitive model design |
| Normal distribution | • Simple implementation | • Does not capture the skewness of the data<br>• Risk of truncation at zero for high driving intensities |
| Weibull distribution | • Captures the skewness of the data and accounts for longer tails<br>• Distribution is by definition always larger than zero | • Shape parameter of the Weibull distribution is more difficult to interpret<br>• May overestimate longer tails |

Supplementary Fig. 6. The elasticity design instead results in vehicle lifetimes that approach zero for very high driving intensities.

Regarding the choice of distribution, the Weibull distribution benefits from better reflecting the skewness of the statistics. However, it overcompensates for higher driving intensities when applied with the logistic design, resulting in longer tails of vehicle lifetimes than the statistics indicate, see the greater distance between lines in Fig. 4b. This difference between Normal- and Weibull-based model designs is close to negligible for the elasticity design. Benefits and drawbacks for the choice of distribution and model design are summarized in Table 1.

**Prospective lifecycle assessment with vehicle fleet turnover**

A vehicle fleet turnover simulation is designed to evaluate the impact of lifetime-intensity elasticities on the carbon footprint for individually owned BEVs, individually owned autonomous BEVs and shared autonomous BEVs. The simulations also test assumptions on how many individual owned BEVs that a shared autonomous BEV can replace, and different levels of implied empty travel of shared autonomous BEVs.

Average carbon footprints (reported in g $CO_2$ per vehicle-kilometer of intended travel) are estimated for the fleet using a prospective lifecycle assessment framework based on GREET® 2 - Version 2019[55] adapted for scenario analysis[4]. The framework enables estimations of future carbon footprints of passenger cars depending on climate change mitigation efforts in electricity generation and global manufacturing. Two pathways for this mitigation are analyzed: one in line with stated policies and one that achieves the goals of the Paris Agreement. The results presented in the main paper assumes a pathway that achieves the goals of the Paris Agreement, while the results for stated policies are presented in the Supplementary Information. The stated policies pathway is based on currently implemented and stated climate policies by 2019 and the pathway in line with the goals of the Paris Agreement is designed to limit global mean temperature increase to below 1.8 °C. The two pathways are based on the IEA[56] scenarios named Stated Policies and Sustainable Development.

The vehicle fleet turnover simulation is designed for the fleet to match a constant annual travel demand equal to $N_O = 1000$ cars driving at average driving intensity, $D_O = 14{,}200$ km per year. For each year, $t$, the number of new cars needed are estimated by solving Eqs. (7), (8):

$$S = \hat{S}(t) + N(t) \cdot m\left(\tilde{t} = 0\right), \tag{7}$$

and

$$S = N_0 \cdot D_0, \tag{8}$$

where the number of new cars, $N$, is estimated as the difference between the annual travel demand and the annual travel range of the current fleet, $\hat{S}$, divided by the annual vehicle-kilometers, $m$, for a new car.

The travel range of the current fleet, $\hat{S}$, in each year, $t$, is given by

$$\hat{S}(t) = \sum_{\tilde{t}} \left( \hat{S}\left(t-1,\tilde{t}\right) - \Phi_W\left(t,\tilde{t},D\right) \right) \cdot m\left(\tilde{t}\right), \tag{9}$$

where the fleet of the previous year is an age distribution for age cohorts, $\tilde{t}$, from age 1 to 40 years, in one-year steps. The initial age distribution for the first year is estimated by a Weibull distribution (shape 1.4 and scale 13). The distribution is informed by statistics on the age of the Swedish vehicle car fleet[57] and serves to initiate the simulation, which is run for 50 iterations (years) to give it time to stabilize at a steady state level. The annual range covered by a car of a given age, $\tilde{t}$, is given by

$$m\left(\tilde{t}\right) = \frac{D_0 \cdot \tau_0}{\sum_{\hat{t}=0}^{\tau_0} (1-b)^{\hat{t}}} \cdot (1-b)^{\tilde{t}}, \tag{10}$$

where the annual driving range is assumed to decrease by $b = 4.4\%$ per year over its lifetime (estimated based on statistics on driving distances in Sweden[57]), $D_O$ represents the average annual driving intensity and $\tau_0$ represents the mean vehicle lifetime. The retirements for each age cohort in year, $t$, are given by the cumulative probability distribution for the semi-empirical lifetime-intensity model, assuming the elasticity design and Weibull distribution, as described in Eqs. (2), (3), (6), assuming that the driving intensity, $D$, is equal to $D_i \cdot (1 + \theta)$, where $D_i$ is the intended travel distance and $\theta$ represents the additional share of empty travel. The probability of retirement earlier than a lifetime of one year is added to the probability of retirement at the one-year mark. This is to avoid truncating the probabilities for retirement for cars with a lifetime of less than one year, which is a risk for the extreme case of ($\varepsilon = -1$).

The model returns annual sales, stock, and retirements. Carbon footprints per km, CF, associated with that steady state are estimated based on the total manufacturing- and use-phase emissions, $E$, for each year divided by the total intended traveling distance, $N_0 \cdot D_0$:

$$CF(t) = \frac{E_{\text{Manufacturing}}(N(t),t) + E_{\text{Use}}(S \cdot (1+\theta), t)}{N_0 \cdot D_0} \tag{11}$$

Manufacturing-phase $CO_2$ emissions, $E_{\text{Manufacturing}}$, are estimated for car sales in each year based on manufacturing processes as implemented in GREET® for the Stated Policies Scenario, while new and innovative processes are phased in over time for the Sustainable Development Scenario based on a literature review[4]. Use-phase $CO_2$ emissions, $E_{\text{Use}}$, are estimated annually based on total traveled distance (including potential empty travel), vehicle energy use, and appropriate carbon intensities described below. The specific energy use of the cars are assumed be 201 Wh per km[4]. Autonomous BEVs are assumed to have the same specific energy use per km as regular BEVs. BEVs are assumed to charge with electricity produced using average global, European, or Swedish technology mixes (results for European and Swedish technology mixes are presented in the Supplementary Information).

The carbon intensity of electricity is based on estimates of average direct emissions for future electricity mixes for each respective geographic area, see description of the data sources for the scenarios below. 2019 is used as a base year to avoid the influence of the Covid-pandemic on the carbon intensities. The carbon intensities used for

electricity represent averages for each respective geographic area following the attributional nature of the chosen prospective lifecycle assessment framework[58,59]. Upstream emissions occurring in production of fuels and power stations are accounted for by adding a weighted factor for future electricity mixes based on estimates by Pehl et al.[60]. We assume that Pehl et al.'s estimates of upstream emissions for each electricity generation technology can be applied regardless of geographic area and that their baseline and climate policy scenarios resemble the Stated Policies and Sustainable Development scenarios used in this study. Note that emissions for the construction of water and nuclear power stations are assumed to be zero for Sweden and the European Union due to their long lifetime, the fact that they were mainly constructed several decades ago, and that few new stations are planned. Hence, we assume that the emissions from the construction of these stations are only attributed to electricity production prior to 2019. Continuing to account for these construction-related emissions in the carbon intensity of electricity after 2019 would not have any significant impact on the results.

For the global electricity mix used in manufacturing and for charging, future direct emissions and adjustments to account for transmission and distribution losses (based on the difference between estimated supply and demand) are based on estimates by the IEA[56] for the two decarbonization pathways, Stated Policies and Sustainable Development. For the European electricity mix used for charging, direct emissions and adjustments to account for transmission and distribution losses are based on scenarios by the European Commission[61] combined with the cap of the European Union emissions trading system reaching zero in 2058[62] for both decarbonization pathways. For the Swedish electricity mix used for charging, direct emissions for 2019 are calculated based on the total emissions for electricity generation divided by the end-use of electricity[63,64]. Direct emissions are assumed to decrease linearly to zero by 2045 for both decarbonization pathways, in line with the adopted net-zero emission target and the Swedish government's intention[65] to reach zero emissions from electricity generation. Upstream emissions are based on estimates by Pehl et al.[60] combined with projections for the future electricity generation mix by the IEA[56], European Commission[61], and Swedish Energy Agency[66].

### Reporting summary
Further information on research design is available in the Nature Research Reporting Summary linked to this article.

## Data availability
Data for all figures and additional data used in the analyses are available from the corresponding author upon request. Note that the detailed data on vehicle retirement are treated as confidential since data that could be traced back to individuals or companies are protection under the Swedish Public Access to Information and Secrecy Act (SFS 2009:400). Hence, requests for access to these detailed data should be made directly to the Swedish governmental agency Transport Analysis (https://www.trafa.se/vagtrafik/fordon/ - dataset "Fordon på väg").

## Code availability
The computer code used to generate the results reported in this study are available from the corresponding author upon request.

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

## Acknowledgements

We acknowledge the support for this research by Mistra Carbon Exit financed by Mistra, the Swedish Foundation for Strategic Environmental Research (J.M. and D.J.A.J.), and by the NAVIGATE project financed by the European Commission, H2020/2019-2023, grant agreement number 821124 (D.J.A.J.).

## Author contributions

J.M. and D.J.A.J. contributed jointly to conceptualizing the research and writing the article. J.M. performed the statistical analysis and computational studies.

## Funding

## Competing interests

The authors declare no competing interests.
