## [Peer Review File · Nature Communications]

Impacts of shared mobility on vehicle lifetimes and the carbon footprint of electric vehiclesReviewers' Comments:

Reviewer #1:

Remarks to the Author:

Dear authors,

This is a very interesting study investigating the relationship between vehicle use intensity, lifetime, the implications of shared mobility, and the carbon footprint. I think there is potential for this piece of study to be accepted for publication. But intensive revisions are needed.

- Generally, this study seems very like a combination of three pieces of small studies, which are (1) the relationship between vehicle use intensity, lifetime and lifetime driving distance, which is obtained by using conventional ICEV as the example. (2) a carbon footprint calculation for BEV with the previous relationships in (1) incorporated. (3) a calculation on the empty travel that can be tolerated. I would say these three parts of the study are not very closely integrated with each other. It is more like a loose combination of researches in three different research areas, and trying to find some stories behind the combination. I recommend that the authors might put far more focus on the first part of the study, because currently there lacks understanding in the relationship between driving intensity and vehicle lifetime. And the first part could be not necessarily connected to the BEV carbon footprint and shared AV empty mileage contexts.

- Another problem when the authors try to incorporate the relationship found in conventional vehicles to shared AVs is that as people are expecting longer driving range for AVs, so the lifetime design for AVs could be very different from conventional vehicles. In that circumstance, I am afraid that the analysis could incorporate significant bias.

- Figure 1, I would recommend the authors to expand the explanations on this figure. More information should be provided, for example, the relationship between lifetime and lifetime driving distance; driving distance and lifetime driving distance.

- To use the relationship obtained on ICEVs on BEVs could incorporate biases, so that I would recommend the authors to conducted some sensitivity analysis.

- If the authors have already determine the value of e to be -0.59 , does it make a lot of senses to analyze the cases of e equaling to 0 and 1?

- the empty travel section, all the discussions are very hard to follow due to the complication to the analysis. I recommend the authors to simply the discussion and highlight the most important findings in an easy-to-follow way.

- Line 241, yes higher driving intensity leads to shorter vehicle lifetime, which makes the use-phase GHG emissions because more emissions occur in the near future than in the long future. However, this analysis is only from one-vehicle perspective. But when considering that after the retirement of this vehicle, the next vehicle will get the benefit of lower emissions of electricity in the long future, than the near-future loss could be filled.

- line 243, the analysis on the manufacturing-phase GHG emissions actually need also to incorporate the impact from electricity emission factor, as much of energy consumption for battery production is electricity. This will lead to some differences in the manufacturing phase emissions, like the impact from near-term emission factor and long-term emissions factor analyzed in the use-phase emissions.

- line 257, the authors should not only identify the research weakness, actually this can be addressed by a sensitivity analysis, which I recommend that the authors should add.

Reviewer #3:

Remarks to the Author:

Summary of contribution:

This paper contributes meaningfully to the literature on vehicle lifetime carbon footprints by using retirement statistics to model relationships between driving intensity and lifetime, and by using those relationships to characterize whether high driving intensities tend to improve or harm carbon footprints. Sensitivity to an appropriate range of models is considered. The value of these findings largely comes from its implications for "future mobility", e.g., ridesourcing, car sharing, and autonomous vehicles. The dataset used is generally appropriate to investigate those implications and provides a much-needed look at intensity-lifetime relationships.

That said, it isn't clear to me whether the dataset can yield findings that cleanly extrapolate to those "future mobility" options -- this may be something that simply warrants an additional caveat in the discussion section but seems important to acknowledge. It also isn't clear to me whether there are specific insights that could translate these vehicle-level findings to fleet-level findings (are fleets of very few high-use cars better or worse than fleets of very many low-use cars?) -- maybe nothing can be said on that front, but maybe there are at least some specific directions the authors can suggest as future work based on their model findings.

Comments on data and methods:

1. The study depends on using historical vehicle retirements to infer trends for future (shared) mobility. It seems like one missing piece of the discussion section is whether this subsetted excerpt of all passenger cars may or may not cleanly extrapolate to uses such as car sharing and ride sourcing. Should we expect that the vehicle failure causes and reasons for retirement will be pretty much the same for shared mobility fleets as for all private vehicles? Or is there anything about their driving cycles (mostly long-duration urban driving shifts), ongoing maintenance regimes, or scrappage decisions (made by professional fleet managers instead of individual owners) that might make the data not extrapolate as cleanly such that results are biased in a direction we can characterize? This may be worth some brief comment.
2. I am glad to see multiple statistical model types tested for the semi-empirical model. The specific tradeoffs between model types (Table 1) are outside of my expertise, but the explanations provided were clear and concise. Including elasticities of 0 and -1 in the main figures helped me understand how the model is working and also serves as a useful bound.
3. This analysis is conducted and presented on a per-vehicle basis. I might hesitate to draw fleetwide conclusions from this analysis, and I think it should be mentioned in the problem framing or in discussion that the interactions between average vehicle use level and fleetwide impacts are outside of the paper's scope. For example, if demand is fixed and served 100% via AVs and ridesourcing, each unit increase in vehicle driving intensity seems to imply fewer total cars are needed -- but I don't think this analysis can tell us whether the net carbon impacts of that increase in intensity would be positive or negative.

Additional comments:

1. Please comment on the choice to use average electricity grid emissions factors instead of marginal and how this choice may affect your results and findings. In particular, as renewables increase, average electricity emissions factors will fall, but marginal emissions factors may or may not change (in nearly all regions the marginal generator is some form of fossil fuel generation). Might using marginal factors alter any findings? If so, it is worth some justification in the text of this choice.
2. It seems that AVs, ridesourcing, and/or car sharing may cause not only empty vehicle travel, but also additional induced demand (ie, the reduced costs or inconveniences of travel due to these additional options may lead to new trip generation). An interesting complement to the sensitivity analysis of breakeven points for deadheading would be a similar look at breakevens for percent of increased vehicle-distance traveled due to new travel demand. This may be out of scope for this

manuscript, but it may not require any new analysis, but instead a re-interpretation of the existing breakeven results. If nothing else, it is worth mentioning in the discussion section that induced demand is a similar issue of concern (conceivably on the same order of magnitude as deadheading in the long term, e.g., if AVs lead to land use changes and drastic shifts in behavior).

3. Enough observations were removed (>50%) that I would hesitate to characterize the exercise as data cleaning -- a more generic term such as filtering, subsetting, or preparation may be more appropriate. This may be pedantic, but given the "cleaning" term, I thought that only anomalous observations were being removed, but in this case, the goal is not simply to remove anomalies but also to create a dataset more representative and applicable to the analysis.

4. Removal of 360,000 vehicles sitting unused for >14 months prior to scrappage: I am not expert in scrappage issues, but removing this many observations (nearly as many as the size of the final dataset) seems to warrant a brief discussion of how we expect it to de-bias the dataset and how including it could have altered findings. (If sitting unused in a garage is how most cars ultimately transition towards scrappage, does excluding it alter findings? Similarly, if many Swedish cars find a second life in other countries but are not included in the data excerpt from the government, would that dataset bias alter findings?)

5. The methods section states that: "The annual driving distance, d , for year t , is assumed to decrease by $b = 4.4\%$ per year". How was this value chosen and are results very sensitive to it?

6. I don't have a specific suggestion, but on plots where Normal distribution and Weibull distribution results are plotted as separate line types (ex: Fig. 4), I found it difficult to make out some of the dotted lines that were close together.

REVIEWER COMMENTS

We would like to thank both reviewers for taking the time to thoroughly review our study. Their comments have greatly improved the manuscript. Please find our point-by-point response to all the comments below. Changes have been made in the corresponding areas in the manuscript. All line references are in regards to the revised manuscript. Text highlighted in yellow in citations show what part of the text that was revised.

Reviewer #1 (Remarks to the Author):

Dear authors,

This is a very interesting study investigating the relationship between vehicle use intensity, lifetime, the implications of shared mobility, and the carbon footprint. I think there is potential for this piece of study to be accepted for publication. But intensive revisions are needed.

- Generally, this study seems very like a combination of three pieces of small studies, which are (1) the relationship between vehicle use intensity, lifetime and lifetime driving distance, which is obtained by using conventional ICEV as the example. (2) a carbon footprint calculation for BEV with the previous relationships in (1) incorporated. (3) a calculation on the empty travel that can be tolerated. I would say these three parts of the study are not very closely integrated with each other. It is more like a loose combination of researches in three different research areas, and trying to find some stories behind the combination. I recommend that the authors might put far more focus on the first part of the study, because currently there lacks understanding in the relationship between driving intensity and vehicle lifetime. And the first part could be not necessarily connected to the BEV carbon footprint and shared AV empty mileage contexts.

Thank you for pointing out that the three parts do not seem closely integrated. We have thoroughly revised the manuscript to improve the coherence of the study and, also added a more thorough analysis of the relationship between driving intensity and vehicle lifetime.

We agree that there is a lack of understanding of the relationship between driving intensity and vehicle lifetime. At the same time, there is a strong trend towards electrification both by many car manufacturers and through the ICEV phase-out policies proposed by several governments worldwide. Furthermore, car sharing and ride sharing are to some extent already implemented (e.g., ride hailing) and thoroughly discussed in academia as well as industry, including in the latest assessment from WG3 of the IPCC. Hence, a study trying to understand the relationship between driving intensity and vehicle lifetime to inform studies of future scenarios for transportation and policy analyses should relate them to the potential issues with electrification and potential future shared AVs. That is the reasoning behind our study's objective and why the three different parts are all vital to the study.

- Another problem when the authors try to incorporate the relationship found in conventional vehicles to shared AVs is that as people are expecting longer driving range for AVs, so the lifetime design for AVs could be very different from conventional vehicles. In that circumstance, I am afraid that the analysis could incorporate significant bias.

Thank you for highlighting this problem. We acknowledge the risk of biases when using data for ICEVs as the basis for discussing the relationship between driving intensity and vehicle lifetime. Although we agree that there is a risk with using data based on ICEVs to analyze the impact of an emerging technology, we argue that both the design of future regular/autonomous BEVs and to what extent longer driving ranges would affect the relationship are highly uncertain. Even if enough data for a statistical analysis of EVs were available, the risk of bias would still persist. To highlight this, the following discussions has been added in the first section of the manuscript (lines 134-167):

"Currently, battery degradation is often raised as a constrain to the cumulative driving distance and lifetime of BEVs²⁸⁻³⁰, but the BEV is a relatively new technology on the market and, hence, statistics on battery lifetimes from real-world driving are scarce. The number of electric vehicles on the world's roads were in the thousands in 2010 and grew rapidly to reach about 2 million by 2016 and over 10 million by 2020^{31,32}. Hence, if enough retirement statistics for electric vehicles were available to make thorough statistical analyses, most vehicles would be much less than 10 years old. However, the limited data currently available on cars with batteries in Swedish vehicle retirement statistics show similar distributions as the stratified data presented above, see Supplementary Notes 1-3 and Supplementary Figures 11-12. However, the data show shorter lifetimes on average (due to the limited historic data on electrified vehicles) and with a bias towards hybrid electric vehicles (HEVs) due to very few BEVs and plug-in hybrid electric vehicles (PHEVs) having been retired during the analyzed period. Many BEV manufacturers already have warranties for their batteries of about seven to eight years or about 150,000 to 240,000 km, whichever comes first³³⁻³⁷. Future battery chemistries may further reduce degradation. Some studies suggest that future batteries may have significantly longer lifetimes than today

through completely different battery chemistries³⁸, changes in charging and use behavior³⁹, and/or changed battery design⁴⁰ that could potentially yield a cumulative driving distance of more than three million kilometers – effectively outliving the vehicle. These improvements, if they materialize, would likely improve the cycling of the batteries. However, other factors could still limit the vehicle's lifetime²⁵, such as accidents, aging of other vehicle parts (e.g., structural elements of chassis and body), economic reasons and consumer trends. Further, the durability of the vehicle is significantly dependent on the vehicle design, material selection and business models⁴¹.

In summary, the results suggest that the annual driving intensity indeed has a strong influence on vehicle lifetimes. The relationship between driving intensity and vehicle lifetime may differ between BEVs and ICEVs, but not enough data is yet available to make such a claim. As a consequence, the remainder of this article explores how changes in annual driving intensity may influence the carbon footprint of passenger car travel, assuming that the relationship shown for ICEVs is applicable as a proxy for individually owned and shared autonomous BEVs. We capture the uncertainty in future vehicle lifetimes of (shared and autonomous) BEVs by highlighting extreme values for the relationship between annual driving intensity and vehicles lifetime as well as the empirically estimated relationship based on ICEV retirement data.”

And the following in the discussion section (lines 376-387):

“Finally, our conclusions rely on the assumption that the relationship between driving intensity and vehicle lifetime established in the semi-empirical model will hold also for future regular and autonomous BEVs. In this article, we present preliminary evidence suggesting that cars with batteries follow similar trends as ICEVs, but the design and use of future batteries and vehicles are still highly uncertain. Hence, the intention here is to highlight potential consequences based on currently available data and discuss them in relation to extreme cases. Those extreme cases highlight a range of plausible outcomes if the lifetime characteristics of future batteries and vehicles may deviate from those of current passenger cars. In any case, the analysis shows that the carbon footprint may be substantially reduced if the relationship between average annual driving intensity and vehicle lifetime is weakened, pointing to the importance of designing future BEVs (both autonomous and regular) for durability.”

- Figure 1, I would recommend the authors to expand the explanations on this figure. More information should be provided, for example, the relationship between lifetime and lifetime driving distance; driving distance and lifetime driving distance.

Thank you for this suggestion. Figure 1 has been revised to visualize not only the relationship between driving intensity and vehicle lifetime but also vehicle lifetime vs. total driving distance and total driving distance vs driving intensity. The following analysis is also added (lines 102-133:

“The stratification is made for individual average annual driving intensity classes, varying from 0 to 100,000 km per year in steps of 10,000 km per year. For each individual driving intensity class, a close to linear relationship exists between vehicle lifetime and cumulative driving distance. The linear slope becomes steeper with each higher driving intensity class, see Figure 1a. This suggests that the calendar age of a vehicle becomes generally shorter with increasing annual driving intensity. Further, the cumulative driving distances are distributed across a wide range for higher driving intensity classes, see Figure 1c, while the distribution is narrower for lower driving intensities. Hence, the probability of a retirement decision at a specific cumulative driving distance becomes smaller as the annual driving intensity increases. A fixed cumulative driving distance is assumed in many lifecycle assessments of vehicles^{13,18}. However, this assumption is not corroborated by the data presented here. Finally, the distribution of vehicle lifetimes becomes narrower and shifts towards lower vehicle lifetimes as the average driving intensity increases, see Figure 1b. Hence, we focus the following analysis on empirically describing the relationship between driving intensity and vehicle lifetime in order to capture the impact of vehicle use on retirement age.

The average vehicle lifetime decreases with each higher driving intensity class, from 19 years for average driving intensities of 0-10,000 km per year to 3.9 years for average driving intensities of 90,001-100,000 km per year, see Figure 1b. The standard deviation of the distributions also indicates that the range of probable lifetimes becomes narrower with increasing annual driving intensity (although the standard deviation increases in relative terms). The standard deviation decreases from 5.0 years for driving intensities of 0-10,000 km per year to 1.9 years for driving intensities of 90,001-100,000 km per year (assuming Normal-distributed data). Results for a categorization in four vehicle sizes (mini, medium, large and luxury size cars, see Supplementary Figure 5) suggest that cars with low annual driving intensity are mainly represented by small size cars, while large to luxury size cars mainly have higher annual driving intensities. Medium size cars cover the full spectrum of annual driving intensities.”

- To use the relationship obtained on ICEVs on BEVs could incorporate biases, so that I would recommend the authors to conducted some sensitivity analysis.

- If the authors have already determine the value of e to be -0.59, does it make a lot of senses to analyze the

cases of ϵ equaling to 0 and 1?

Thanks again for highlighting the risk of bias in the dataset. We have added a visualization (Supplementary Figure 11) of the data points available for cars with batteries in the analyzed dataset. Although the dataset is too limited for a thorough statistical analysis, the available data points follow similar distributions as the data analyzed for ICEVs. This is thoroughly discussed in Supplementary Notes 1-3. The following discussion has also been added to the main body of the manuscript (lines 134-157), as previously mentioned.

"Currently, battery degradation is often raised as a constrain to the cumulative driving distance and lifetime of BEVs²⁸⁻³⁰, but the BEV is a relatively new technology on the market and, hence, statistics on battery lifetimes from real-world driving are scarce. The number of electric vehicles on the world's roads were in the thousands in 2010 and grew rapidly to reach about 2 million by 2016 and over 10 million by 2020^{31,32}. Hence, if enough retirement statistics for electric vehicles were available to make thorough statistical analyses, most vehicles would be much less than 10 years old. However, the limited data currently available on cars with batteries in Swedish vehicle retirement statistics show similar distributions as the stratified data presented above, see Supplementary Notes 1-3 and Supplementary Figures 11-12. However, the data show shorter lifetimes on average (due to the limited historic data on electrified vehicles) and with a bias towards hybrid electric vehicles (HEVs) due to very few BEVs and plug-in hybrid electric vehicles (PHEVs) having been retired during the analyzed period."

Many BEV manufacturers already have warranties for their batteries of about seven to eight years or about 150,000 to 240,000 km, whichever comes first³³⁻³⁷. Future battery chemistries may further reduce degradation. Some studies suggest that future batteries may have significantly longer lifetimes than today through completely different battery chemistries³⁸, changes in charging and use behavior³⁹, and/or changed battery design⁴⁰ that could potentially yield a cumulative driving distance of more than three million kilometers – effectively outliving the vehicle. These improvements, if they materialize, would likely improve the cycling of the batteries. However, other factors could still limit the vehicle's lifetime²⁵, such as accidents, aging of other vehicle parts (e.g., structural elements of chassis and body), economic reasons and consumer trends. Further, the durability of the vehicle is significantly dependent on the vehicle design, material selection and business models⁴¹."

Finally, we would also like to highlight the benefit of presenting results for elasticities ranging from 0 to -1 as a way of testing the sensitivity in the carbon footprint estimations as well as the empty travel breakeven level. Including these two extreme cases serves two purposes: (i) increasing the understanding of the model design, and (ii) how sensitive the model is to the relationship between driving intensity and vehicle lifetime. Hence, if future shared autonomous BEVs are designed in a way where the driving intensity plays a less important role in the decision to retire vehicles, the results are more likely related to an elasticity close to 0. This could be the case if battery degradation is less influenced by going through many charging cycles. The opposite case, where calendar lifetime plays a less important role in the decision to retire vehicles and the elasticity is close to -1, represents a future where batteries are largely impacted by the number of charging cycles and the total driving distance is fixed. The latter assumption is often used in LCA studies, in which a certain total driving distance over the vehicle's lifetime is assumed. However, the elasticity of -1 case seems less realistic given the incentives for battery manufacturers to improve battery longevity and enable batteries to cope with extreme events such as fast charging and recent laboratory studies supporting that such battery chemistries are feasible (Yang et al., 2021). We highlight this benefit of the extreme cases more clearly in the manuscript (lines 182-194):

"Carbon footprints are also estimated for two extreme cases, $\epsilon = 0$ and $\epsilon = -1$, representing no influence of driving intensity on lifetime and full influence of driving intensity, respectively. The two extreme cases show the sensitivity of the model design to the assumed elasticity. The range represents possible cases if the model was trained on different retirement data, such as future BEVs when sufficient data becomes available. $\epsilon = 0$ is a relevant extreme case if future individually owned and/or shared autonomous BEVs are designed in a way where driving intensity has no importance in the decision to retire vehicles. This could be the case if the vehicle and battery degradation is only influenced by calendar age. $\epsilon = -1$ represent a case where vehicle aging, including aging of the battery, is only dependent on distance driven (i.e., battery aging only depends on the number of charging cycles). This approach is used in many lifecycle assessments^{13,18}, where fixed cumulative vehicles distances are assumed. Note though that the elasticity affecting the distribution is based on the empirical data ($\beta \approx 0.51$) also for the extreme cases."

We also note that Reviewer #3 consider the analysis of the extreme values for the elasticity as a strength of the study.

- the empty travel section, all the discussions are very hard to follow due to the complication to the analysis. I recommend the authors to simply the discussion and highlight the most important findings in an easy-to-follow way.

We are sorry that you found this section hard to follow and agree that it is complex. We have thoroughly reworked this section in the revised manuscript and highlighted the main outcome that now is based on the fleet-wide analysis.

- Line 241, yes higher driving intensity leads to shorter vehicle lifetime, which makes the use-phase GHG emissions because more emissions occur in the near future than in the long future. However, this analysis is only from one-vehicle perspective. But when considering that after the retirement of this vehicle, the next vehicle will get the benefit of lower emissions of electricity in the long future, than the near-future loss could be filled.

Thank you for this suggestion. We agree that modelling a fleet would be a more accurate way of determining the impact of shared autonomous BEVs on the carbon footprint. Hence, we have reworked the sections on the carbon footprint impacts (lines 168-261) and the breakeven level for empty travel (lines 262-339) using a simple vehicle fleet turnover simulation considering a fleet of 1000 vehicles. The carbon footprint estimation section of the Methods section was also revised to include the vehicle fleet turnover simulation (lines 498-588). The carbon footprint estimation section was also moved to the end of the Methods section to follow the structure of the manuscript in general. Since these two sections and the related section in Methods are fully reworked, we have not included the whole text here.

We would also like to highlight that implementing our semi-empirical lifetime-intensity model in a fleet-wide analysis revealed additional aspects of the model that are important to consider when implementing the full distributions. Using the elasticity design with Normal distributions is simple and easy to understand but has a vital flaw when analyzing high driving intensities. As driving intensities increase, the distribution shifts to lower and even negative lifetimes, which of course is not realistic. This issue is overcome by instead using Weibull distributions, which is by definition always larger than zero. Hence, we have changed the elasticities used in the carbon footprint analysis to those estimated for the Weibull distribution and added the caveat of the Normal distribution to Table 1. Again, thank you for suggesting us to go beyond the one-vehicle perspective, which highlighted this important aspect of our lifetime-intensity model design.

- line 243, the analysis on the manufacturing-phase GHG emissions actually need also to incorporate the impact from electricity emission factor, as much of energy consumption for battery production is electricity. This will lead to some differences in the manufacturing phase emissions, like the impact from near-term emission factor and long-term emissions factor analyzed in the use-phase emissions.

Thank you for highlighting that this was not clear in the previous version. The model does incorporate the impact of the electricity emission factor on emissions related to battery production. All vehicles and batteries are assumed to be produced by global average manufacturing industries, using average global electricity. The electricity emission factor is based on direct emissions estimated by the IEA and adjusted to account for upstream processes, as described in the Methods section. The Methods highlights this on the following lines (546-549 and 574-577), included below for your convenience.

“Manufacturing-phase CO₂ emissions are estimated for car sales in each year based on manufacturing processes as implemented in GREET® for the Stated Policies Scenario, while new and innovative processes are phased in over time for the Sustainable Development Scenario based on a literature review⁴.”

“For the global electricity mix used in manufacturing and for charging, future direct emissions and adjustments to account for transmission and distribution losses (based on the difference between estimated supply and demand) are based on estimates by the IEA⁵⁶ for the two decarbonization pathways, Stated Policies and Sustainable Development.”

- line 257, the authors should not only identify the research weakness, actually this can be addressed by a sensitivity analysis, which I recommend that the authors should add.

Thanks again for highlighting these issues. We hope that the analyses and discussions added, and outlined above, are sufficient to address this main weakness of our study.

Reviewer #3 (Remarks to the Author):

Summary of contribution:

This paper contributes meaningfully to the literature on vehicle lifetime carbon footprints by using retirement statistics to model relationships between driving intensity and lifetime, and by using those relationships to characterize whether high driving intensities tend to improve or harm carbon footprints. Sensitivity to an appropriate range of models is considered. The value of these findings largely comes from its implications for "future mobility", e.g., ridesourcing, car sharing, and autonomous vehicles. The dataset used is generally appropriate to investigate those implications and provides a much-needed look at intensity-lifetime relationships.

That said, it isn't clear to me whether the dataset can yield findings that cleanly extrapolate to those "future mobility" options -- this may be something that simply warrants an additional caveat in the discussion section but seems important to acknowledge. It also isn't clear to me whether there are specific insights that could translate these vehicle-level findings to fleet-level findings (are fleets of very few high-use cars better or worse than fleets of very many low-use cars?) -- maybe nothing can be said on that front, but maybe there are at least some specific directions the authors can suggest as future work based on their model findings.

Thank you for taking the time to thoroughly review our manuscript. We have thoroughly revised the manuscript in response to the comments, including replacing the previous carbon footprint analysis with one based on vehicle fleet turnover simulations as well as more thorough discussions on the applicability of the results for future mobility options. The latter also includes insights from the available but limited data on vehicle retirement of cars with batteries.

Comments on data and methods:

1. The study depends on using historical vehicle retirements to infer trends for future (shared) mobility. It seems like one missing piece of the discussion section is whether this subsetted excerpt of all passenger cars may or may not cleanly extrapolate to uses such as car sharing and ride sourcing. Should we expect that the vehicle failure causes and reasons for retirement will be pretty much the same for shared mobility fleets as for all private vehicles? Or is there anything about their driving cycles (mostly long-duration urban driving shifts), ongoing maintenance regimes, or scrappage decisions (made by professional fleet managers instead of individual owners) that might make the data not extrapolate as cleanly such that results are biased in a direction we can characterize? This may be worth some brief comment.

Thank you for highlighting this problem. We acknowledge the risk of biases when using data for ICEVs as the basis for discussing the relationship between driving intensity and vehicle lifetime. Although we agree that there is a risk with using data based on ICEVs to analyze the impact of an emerging technology, we argue that both the design of future regular/autonomous BEVs and to what extent longer driving ranges would affect the relationship are highly uncertain. Even if enough data for a statistical analysis of EVs were available, the risk of bias would still persist. To highlight this, the following discussions has been added in the first section of the manuscript (lines 134-167):

"Currently, battery degradation is often raised as a constrain to the cumulative driving distance and lifetime of BEVs²⁸⁻³⁰, but the BEV is a relatively new technology on the market and, hence, statistics on battery lifetimes from real-world driving are scarce. The number of electric vehicles on the world's roads were in the thousands in 2010 and grew rapidly to reach about 2 million by 2016 and over 10 million by 2020^{31,32}. Hence, if enough retirement statistics for electric vehicles were available to make thorough statistical analyses, most vehicles would be much less than 10 years old. However, the limited data currently available on cars with batteries in Swedish vehicle retirement statistics show similar distributions as the stratified data presented above, see Supplementary Notes 1-3 and Supplementary Figures 11-12. However, the data show shorter lifetimes on average (due to the limited historic data on electrified vehicles) and with a bias towards hybrid electric vehicles (HEVs) due to very few BEVs and plug-in hybrid electric vehicles (PHEVs) having been retired during the analyzed period.

Many BEV manufacturers already have warranties for their batteries of about seven to eight years or about 150,000 to 240,000 km, whichever comes first³³⁻³⁷. Future battery chemistries may further reduce degradation. Some studies suggest that future batteries may have significantly longer lifetimes than today through completely different battery chemistries³⁸, changes in charging and use behavior³⁹, and/or changed battery design⁴⁰ that could potentially yield a cumulative driving distance of more than three million kilometers – effectively outliving the vehicle. These improvements, if they materialize, would likely improve the cycling of the batteries. However, other factors could still limit the vehicle's lifetime²⁵, such as accidents, aging of other vehicle parts (e.g., structural elements of chassis and body), economic reasons and consumer trends. Further, the durability of the vehicle is significantly dependent on the vehicle design, material selection and business models⁴¹.

In summary, the results suggest that the annual driving intensity indeed has a strong influence on vehicle lifetimes. The relationship between driving intensity and vehicle lifetime may differ between BEVs and ICEVs, but not enough data is yet available to make such a claim. As a consequence, the remainder of this article explores how changes in annual driving intensity may influence the carbon footprint of passenger car travel, assuming that the relationship shown for ICEVs is applicable as a proxy for individually owned and shared autonomous BEVs. We capture the uncertainty in future vehicle lifetimes of (shared and autonomous) BEVs by highlighting extreme values for the relationship between annual driving intensity and vehicles lifetime as well as the empirically estimated relationship based on ICEV retirement data."

And the following in the discussion section (lines 376-387):

“Finally, our conclusions rely on the **assumption that the** relationship between driving intensity and vehicle lifetime established in the semi-empirical model will hold also for future **regular and autonomous BEVs**. **In this article, we present preliminary evidence suggesting that cars with batteries follow similar trends as ICEVs, but the design and use of future batteries and vehicles are still highly uncertain. Hence, the intention here is to highlight potential consequences based on currently available data and discuss them in relation to extreme cases. Those extreme cases highlight a range of plausible outcomes if the lifetime characteristics of future batteries and vehicles may deviate from those of current passenger cars. In any case, the analysis shows that the carbon footprint may be substantially reduced if the relationship between average annual driving intensity and vehicle lifetime is weakened, pointing to the importance of designing future BEVs (both autonomous and regular) for durability.**”

2. I am glad to see multiple statistical model types tested for the semi-empirical model. The specific tradeoffs between model types (Table 1) are outside of my expertise, but the explanations provided were clear and concise. Including elasticities of 0 and -1 in the main figures helped me understand how the model is working and also serves as a useful bound.

Thank you! We agree that highlighting the extreme values for the elasticities in the figures improves understanding of the model and how sensitive it is to the relationship between driving intensity and vehicle lifetime. Hence, if future shared AVs are designed in a way where the driving intensity plays a less important role in the decision to retire vehicles, the results are more likely related to an elasticity close to 0. This could be the case if battery degradation is less influenced by going through many charging cycles. The opposite case, where vehicle lifetime plays a less important role in the decision to retire vehicles and the elasticity is close to -1, represents a future where batteries are largely impacted by the number of charging cycles.

We have highlighted this benefit more clearly in the manuscript (lines 182-194):

“Carbon footprints are also estimated for two extreme cases, $\epsilon = 0$ and $\epsilon = -1$, representing no influence of driving intensity on lifetime and full influence of driving intensity, respectively. **The two extreme cases show the sensitivity of the model design to the assumed elasticity. The range represents possible cases if the model was trained on different retirement data, such as future BEVs when sufficient data becomes available. $\epsilon = 0$ is a relevant extreme case if future individually owned and/or shared autonomous BEVs are designed in a way where driving intensity has no importance in the decision to retire vehicles. This could be the case if the vehicle and battery degradation is only influenced by calendar age. $\epsilon = -1$ represent a case where vehicle aging, including aging of the battery, is only dependent on distance driven (i.e., battery aging only depends on the number of charging cycles). This approach is used in many lifecycle assessments^{13,18}, where fixed cumulative vehicles distances are assumed. Note though that the elasticity affecting the distribution is based on the empirical data ($\beta \approx 0.51$) also for the extreme cases.**”

3. This analysis is conducted and presented on a per-vehicle basis. I might hesitate to draw fleetwide conclusions from this analysis, and I think it should be mentioned in the problem framing or in discussion that the interactions between average vehicle use level and fleetwide impacts are outside of the paper's scope. For example, if demand is fixed and served 100% via AVs and ridesourcing, each unit increase in vehicle driving intensity seems to imply fewer total cars are needed -- but I don't think this analysis can tell us whether the net carbon impacts of that increase in intensity would be positive or negative.

Thank you for this suggestion. We agree that modelling a fleet would be a more accurate way of determining the impact of shared autonomous BEVs on the carbon footprint. Hence, we have reworked the sections on the carbon footprint impacts (lines 168-261) and the breakeven level for empty travel (lines 262-339) using a simple vehicle fleet turnover simulation considering a fleet of 1000 vehicles. The carbon footprint estimation section of the Methods section was also revised to include the vehicle fleet turnover simulation (lines 498-588). The carbon footprint estimation section was also moved to the end of the Methods section to follow the structure of the manuscript in general. Since these two sections and the related section in Methods are fully reworked, we have not included the whole text here.

We would also like to highlight that implementing our semi-empirical lifetime-intensity model in a fleet-wide analysis revealed additional aspects of the model that are important to consider when implementing the full distributions. Using the elasticity design with Normal distributions is simple and easy to understand but has a vital flaw when analyzing high driving intensities. As driving intensities increase, the distribution shifts to lower and even negative lifetimes, which of course is not realistic. This issue is overcome by instead using Weibull distributions, which is by definition always larger than zero. Hence, we have changed the elasticities used in the carbon footprint analysis to those estimated for the Weibull distribution and added the caveat of the Normal distribution to Table 1. Again, thank you for suggesting us to go beyond the one-vehicle perspective, which highlighted this important aspect of our lifetime-intensity model design.

Additional comments:

1. Please comment on the choice to use average electricity grid emissions factors instead of marginal and how this choice may affect your results and findings. In particular, as renewables increase, average electricity

emissions factors will fall, but marginal emissions factors may or may not change (in nearly all regions the marginal generator is some form of fossil fuel generation). Might using marginal factors alter any findings? If so, it is worth some justification in the text of this choice.

Marginal coefficients are preferred in consequential LCAs, where the effects on the system of increased use of the analyzed product or service is assessed (Yang, 2016). A prospective LCA can be either consequential or attributional (Arvidsson et al., 2018b), where an attributional prospective LCA provides a snapshot of the future given the evolution of foreground as well as background systems in line with scenario assumptions. A typical consequential prospective LCA would have the intention to go one step further in order to evaluate what impact a specific decision, in our case introduction of shared AVs, would have on each sub-system (Jones et al., 2017), e.g., vehicle manufacturing or electricity generation. This is considered to be beyond the scope of this study. Although, as clearly stated in Arvidsson et al. (2018a): "... an attributional LCA ... [is] effectively identical to a consequential LCA where only first-order (linear) physical flow consequences are considered – or "a consequential LCA based on the attributional [LCA] framework"."

Hence, given the nature of the prospective LCA applied in this study, average coefficients (incl. carbon intensity of electricity) are preferred. Nevertheless, the main concern with applying average carbon intensity of electricity should still be addressed – the risk that changes in electricity demand in response to the analyzed question is large enough to affect the carbon intensity of electricity (Harmsen and Graus, 2013). To handle this risk, care has been taken to choose scenarios for background systems (incl. electricity generation) that match foreground systems. The scenarios for electricity generation referenced for Swedish and EU averages assume increased electricity demand following strong electrification not only in transportation but also in industry. The IEA scenarios (i.e., Stated Policies and Sustainable Development) both also account for increases in electricity demand, specifically looking at global trends within industry and electric vehicle deployment among others.

The following was added to the manuscript in response to this (lines 559-561):

"The carbon intensities used for electricity represent averages for each respective geographic area following the attributional nature of the chosen prospective lifecycle assessment framework^{58,59}."

2. It seems that AVs, ridesourcing, and/or car sharing may cause not only empty vehicle travel, but also additional induced demand (ie, the reduced costs or inconveniences of travel due to these additional options may lead to new trip generation). An interesting complement to the sensitivity analysis of breakeven points for deadheading would be a similar look at breakevens for percent of increased vehicle-distance traveled due to new travel demand. This may be out of scope for this manuscript, but it may not require any new analysis, but instead a re-interpretation of the existing breakeven results. If nothing else, it is worth mentioning in the discussion section that induced demand is a similar issue of concern (conceivably on the same order of magnitude as deadheading in the long term, e.g., if AVs lead to land use changes and drastic shifts in behavior).

Thanks for highlighting the important aspect of induced travel demand for AVs and car sharing and/or ride sharing. We do consider this to be outside of the scope of this particular study and mention this as one of the trends that will affect pathways for decarbonizing passenger car travel, see lines 344-346:

"These trends will affect the pathways towards decarbonization of passenger car travel, including changes in **charging patterns¹³**, cost structures⁹ and the value of travel time⁴²⁻⁴⁴, which may induce additional travel activity⁴⁵ and cause modal shifts^{46,47}."

To further highlight the impact induced travel demand could have, we have added the following sentence in the discussion section (lines 368-375):

"Note that induced travel by autonomous BEVs (both individually owned and shared) has not been assessed in the study. However, this risk is important to consider since the use of autonomous vehicles may substantially increase the travel demand. Autonomous vehicles may effectively reduce the value of travel time and the generalized travel cost⁴⁵ since the driver does not need to be attentive and can instead use their time in the vehicle for whatever they find convenient. This potential increases in the travel demand could further increase the total carbon footprint for the fleet as a whole."

3. Enough observations were removed (>50%) that I would hesitate to characterize the exercise as data cleaning -- a more generic term such as filtering, subsetting, or preparation may be more appropriate. This may be pedantic, but given the "cleaning" term, I thought that only anomalous observations were being removed, but in this case, the goal is not simply to remove anomalies but also to create a dataset more representative and applicable to the analysis.

Thank you for this suggestion. The wording has been changed to "filtering" throughout the manuscript.

4. Removal of 360,000 vehicles sitting unused for >14 months prior to scrappage: I am not expert in scrappage issues, but removing this many observations (nearly as many as the size of the final dataset) seems to warrant a brief discussion of how we expect it to de-bias the dataset and how including it could have altered findings. (If sitting unused in a garage is how most cars ultimately transition towards scrappage, does excluding it alter findings? Similarly, if many Swedish cars find a second life in other countries but are not included in the data excerpt from the government, would that dataset bias alter findings?)

Thanks for raising this point. We have also reconsidered the other filtering categories and criteria used since we realized that we had been too conservative in the set criteria. We have decided to make the following adjustments: the time considered between first registration and the manufacturing year has been increased to two years instead of one (resulting in 107,430 additional vehicles), engine types include also ethanol and natural gas since they are internal combustion engines (15,268 additional vehicles), and the average distance traveled must not be greater than 600 km per day instead of 400 km per day (153 additional vehicles). Note that the number of additional vehicles may not sum to the total since one vehicle may fulfil more than one criterion. This enabled us to do a more even stratification of the data with equal size categories for driving intensity (between 0 and 100,000 km/year in 10,000 km/year steps). The adjustments resulted in a slightly higher elasticity of -0.67 instead of -0.59 (fitted to Normal distribution).

In response to your comment on the time between de-registration and last inspection, we have performed a sensitivity analysis. If the time is increased with two years (i.e., vehicles are removed if they sit unused for > 38 months), 88,568 vehicles are removed instead of 360,110 and the elasticity would be -0.65 (fitted to Normal distribution). Hence, largely unaffected. The following statement was added to the manuscript (lines 408-409):

"Criterion (ii) filters many observations but including them does not significantly impact the results."

Exports were limited 2014-2016 – 8-18 % of total annual de-registrations. For 2017, the share increased to 25% and from 2018 onwards it's been at levels of 35-40%. The number of annually retired vehicles has been fairly constant over the same time period (equal to the time period for our data set – 2014-2018). The exports have been distributed across all vehicle types, but the recent increase in export has been specifically in the age segment 0-5 years. The Swedish governmental agency Traffic Analysis ties this to a recent policy that gives incentives for new car purchases and the leasing deals that lasts around 3 years (see https://www.trafa.se/vagtrafik/export_av_personbilar_okade_kraftigt_2018-8201/, <https://www.trafa.se/vagtrafik/fordon/export-av-personbilar-2020-12094/>, both in Swedish). The fact that the number of retired vehicles has been fairly constant over the time period, also supports that the increase in exports are likely related to the purchasing prerequisites and should not have any major impact on the findings. It is more difficult to draw a conclusion for the exports during 2014-2016, but the impact on the results should in any case be minor since they still are limited in number compared to the full dataset.

5. The methods section states that: "The annual driving distance, d , for year t , is assumed to decrease by $b = 4.4\%$ per year". How was this value chosen and are results very sensitive to it?

Thank you highlighting that we omitted including the source. The value is estimated based on statistics on driving distances in Sweden from the governmental agency Transport Analysis. The source has been added to the list of references and the following is added in the Methods section (lines 532-533):

"...the annual driving range is assumed to decrease by $b = 4.4\%$ per year over its lifetime (estimated based on statistics on driving distances in Sweden⁵⁷)."

We have also included a sensitivity analysis of this assumption as part of the new fleet-wide analysis of the breakeven carbon footprint, which is included in Supplementary Figures 9-10. The sensitivity analysis is summarized in the manuscript as follows:

Lines 238-244:

"... Two other factors also contribute to the decreasing manufacturing-phase emissions as driving intensities increase: the assumed reduction in annual driving intensity for each individual car of 4.4% per year, and the time discretization of one year for the vehicle fleet turnover simulation. The significance of the former factor is tested in Supplementary Figure 9, showing slightly higher carbon footprint when driving intensity is assumed to be constant over each vehicle's lifetime."

Lines 330-335:

"... The sensitivity of the assumed driving intensity decrease rate of 4.4% is also tested, showing higher breakeven levels for the additional empty travel with higher driving intensity decrease rates (i.e., when a larger share of the travel for one vehicle is concentrated to early years in the vehicle's lifetime). Nevertheless, the assumed elasticity in the lifetime-intensity model has a higher impact on the results than the assumed driving intensity decrease rate."

6. *I don't have a specific suggestion, but on plots where Normal distribution and Weibull distribution results are plotted as separate line types (ex: Fig. 4), I found it difficult to make out some of the dotted lines that were close together.*

Thank you for highlighting that this graph is difficult to read. We have adjusted the linetype and opacity of the points to improve readability.

Reviewers' Comments:

Reviewer #1:

Remarks to the Author:

Dear authors,

The response to my comments, and the corresponding revisions are very satisfying. I think the paper has been substantially improved. I have no further comments for the paper.

Reviewer #3:

Remarks to the Author:

The authors have responded comprehensively to my initial feedback. In particular, I appreciate the revisions to the carbon footprint and empty travel breakeven level analyses using a fleet turnover model, which I believe strengthened those sections. I also was glad to see the additional discussion of potential biases regarding BEV implications, the relaxation of some dataset filters, and the additional sensitivity analysis of driving distance decrease rate.

My substantive concerns have been addressed, and so I believe the manuscript is appropriate for publication.